# Oral Cancer Detection using Mobile Vision Technology

Lena D. Swamikannan, Akshay Bhagwan Sonawane, Jay S. Patel, C.S. Mani, Lakshmi Narayana, and Lakshman Tamil

*Abstract*—Oral cancer is the 13th most common cancer, affecting 380,000 people globally. The biggest challenge is that in its initial stage, cancer can go unnoticed until it reaches the most advanced, difficult-to-treat stages. Although a 90% survival rate is assured when diagnosed earlier, early-stage detection requires expensive periodic dental check-ups. Under these circumstances, converting a smartphone into a cancer screening tool has the potential to reduce cancer mortality. Mobile vision technology is a promising platform for early diagnosis of oral cancer. The aim of this research is to develop a telemedical mobile application for Oral Cancer Detection (OCD) using deep learning as a backend. This paper details experiments with various lightweight machine learning architectures, including MobileNetV3Large, which achieved an 84% accuracy, 86% sensitivity and 80% specificity on the test data. By incorporating the machine learning model into the smartphone app, users can capture or upload images for instant or offline screening, ensuring on-device processing that maintains privacy. This application promises to revolutionize oral healthcare accessibility and delivery.

*Index Terms*—oral cancer, deep neural networks, mobile vision technology

## I. INTRODUCTION

According to the World Health Organization [1], nearly 3.5 billion people globally suffer from oral diseases, with three-quarters residing in middle-income countries. Furthermore, projections by the American Cancer Society [2] for 2024 estimate that there will be 58,450 new cases and 12,230 deaths resulting from cancers of the oral cavity and pharynx. Oral cancer can be classified into three stages based on the extent of its spread:

1. Localized: The cancer is confined within the organ of origin with no evidence of spread.

2. Regional: The cancer has spread to nearby structures or lymph nodes.

3. Distant: The cancer has metastasized to distant parts of the body, such as the lungs.

The survival rates for oral cancer, particularly cancers of the lip, tongue, and floor of the mouth, are significantly higher when the disease is detected and treated while still in the localized stage. Many patients when they first notice the lesions (localized stage), tend to dismiss them as normal and overlook them, but these lesions have the potential to turn into cancerous ones.

Regular screening by dentists is essential to enable early diagnosis of this type of cancer. But the high cost of dental insurances and dental office visits are barriers for most of the people to go for regular dental visits. According to the 2023 Gallup poll [3], the proportion of Americans who postponed medical treatment due to cost rose to 38%, up from 26% in 2021. Hence, there is a need to develop screening tools that are cost-effective, easy to use and at the same time are accurate.

The drastic development in camera quality in smartphones has given the power of vision to smartphones. By incorporating medical diagnostic capabilities into smartphones, they can act as personal healthcare assistants. It provides the capability of analyzing the image in real-time using a camera and interpreting the result is termed as Mobile Vision Technology. This technology uses advanced deep-learning models to analyze the image and recognize and/or classify them. The application of Mobile Vision Technology in healthcare offers immense potential for everyone, particularly for those in the lower socio-economic strata and for individuals residing in remote locations away from city centers. We opted to develop the mobile application on the Android platform initially due to its global market dominance and its prevalence among people with limited financial means.

Deep Learning is a boon for medical image analysis, with deep learning-based networks playing a significant role in analyzing images and interpreting results. For accurate diagnosis, these networks should be trained on large datasets. However, our Oral Cancer dataset is small. While there are many histopathological oral cancer datasets available, datasets for smartphone-captured oral cancer images are scarce. The challenge is to develop an efficient deep-learning model for oral cancer screening using a small dataset. To address this challenge, we apply the concepts of data augmentation and transfer learning.

Models that are pre-trained on ImageNet [4] database are used for baseline transfer learning in this research. Though the pre-trained models we've chosen doesn't possess medical domain expertise, it is fine-tuned on our customized Oral database. These models are primarily selected due to their popularity in the world of image classification. The presence of a cancer-specific database could open the door to transformative applications, much like the paradigm-shifting influence of ImageNet in its domain. Just as ImageNet revolutionized general

Lena D. Swamikannan (lxd200013@utdallas.edu), Akshay B. Sonawane (axs180315@utdallas.edu) and Lakshman S. Tamil (laxman@utdallas.edu) are with Erik Jonsson School of Engineering and Computer Science, The University of Texas at Dallas, Richardson, USA.

Jay S. Patel (patel.jay@temple.edu) Department of Oral Health Sciences,Temple University Kornberg School of Dentistry, Philadelphia, USA.

C.S. Mani (dr.cs.mani@gmail.com) and Lakshmi Narayana (drnarayana777@gmail.com) are with Cancer Research and Relief Trust, Chennai, INDIA.

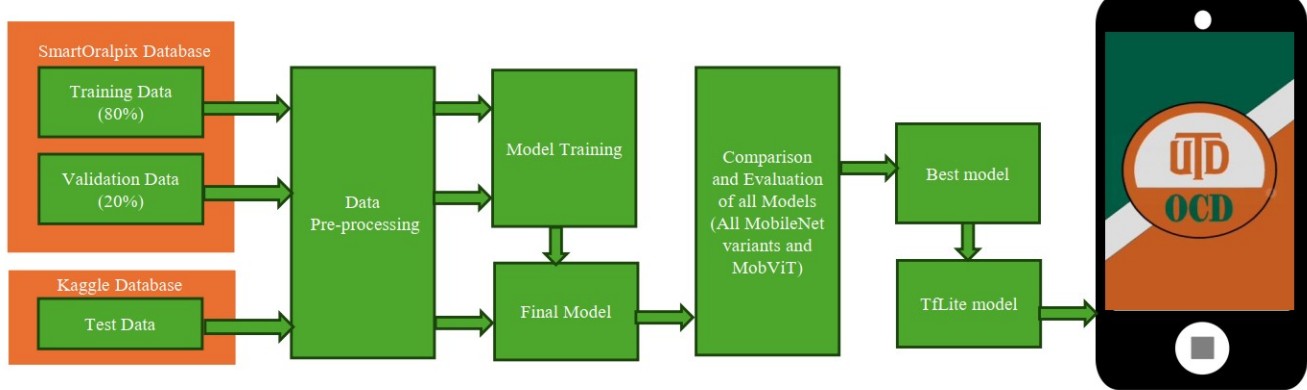

Fig. 1. Workflow for developing a Telemedical Smartphone Application for Oral Cancer Detection: First, we prepare the data according to the specifications of the model to be trained. Then, we perform domain-adaptive fine-tuning on the pre-trained model with the SmartOralpix dataset. Next, evaluations are performed on the test dataset from Kaggle. The above workflow is repeated for all the lightweight models listed in this paper. We pick the best-performing model by analyzing and comparing the results of all the models. Finally, we integrate the best model into an Android application.

image classification, a dedicated cancer-related database holds the potential to drive significant advancements in the realm of cancer image detection and interpretation.

While powerful pre-trained models often achieve higher accuracy, they are not suitable for implementation on mobile phones due to their high computational requirements. Therefore, for our implementation, we have chosen to focus exclusively on lightweight architectures. Specifically, we have experimented with all variants of MobileNet [5], [6], [7] and MobileViT-S [8] due to their design catering to smartphone applications. These models are optimized and then deployed on mobile platforms.

The main contributions of this paper are:

- Developed a customized database of oral images for this research problem.
- Analysed the performance of various lightweight pre-trained models based on CNN and hybrid (CNN+Transformer) approaches, leveraging the combined strengths of local and global processing.
- Proposed transfer learning strategies to fine-tune cross-domain problems.
- Developed an Android app for screening the oral lesions without relying on internet connection, thus increasing accessibility in remote areas.

The rest of the paper is organised as follows: Section II discusses the literature survey performed for this research. Section III introduces the database and its details. Section IV presents the proposed approach to create an efficient model for oral lesion classification. Section V gives the results and discussion of the experiments. Section VI explains the outline of Android app and Section VII reports the conclusions.

## II. RELATED WORK

The computer vision community has extensively studied methods to automate oral cancer detection, where most of these studies are performed on memory-intensive Deep Convolutional Neural Network (DCNN) architectures. There are various types of oral images utilized in diagnosing oral cancer using deep learning models. They are Histopathological images (obtained from examining biopsy samples), Hyperspectral images (analysing spectrum of the image), Computed Tomography (using X-rays) scan images, Autofluorescence images (using special light source usually ultraviolet or bluelight), White light images (using visible light), and Color images (using camera). Research efforts aimed at classifying oral cancer, regardless of the type of image used, have predominantly focused on either traditional CNN models [9]–[19] or Conventional Vision Transformers [20]. Hybrid approaches that combine these methods have not been extensively explored. The studies [14]–[18] did not use pre-trained models, instead they use conventional CNN architecture. Table I summarizes the literature survey on deep learning based studies on Oral Cancer Detection.

### A. Heavyweight Architectures

Zubair et al. [19] made use of AlexNet, GoogleNet, VGG19, Inceptionv3, ResNet50, and SqueezeNet for binary classification (normal and cancer) and multiclass classification (Oral Thrush (OT), Fissured Tongue (FT), Geographic Tongue (GT), Strawberry Tongue (ST), and Leukoplakia (LP)) of oral cancer dataset. This research is focused on lesions occurring in the tongue. Due to limited data availability for the target task, transfer learning concept was applied, in which these networks were pre-trained with ImageNet database. In this work, VGG19 achieved an accuracy of 97.5% for binary classification and ResNet50 achieved an accuracy of 96% for multi-class classification. Sang et al. [21] explored three variants of VGG architecture (VGG-16, VGG-CNN, and VGG-CNN-S). Among these, VGG-CNN-S reported an accuracy of 86.9%. This classifier uses autofluorescence and white light images as the input for the deep learning model. Welikala's

TABLE I
SUMMARY OF RESEARCH STUDIES IN AUTOMATED ORAL CANCER DETECTION

| Reference | Imaging Technique | Architecture Explored | Database Details | Source |
|---|---|---|---|---|
| [20] | Histopathological | Vision Transformer, Xception, ResNet50, InceptionV3, Densenet121, Densenet169, InceptionResNetV2, Densenet201, EfficientNetB7 | Normal-2435, Cancer-2511 | Kaggle |
| [9] | Histopathological | AlexNet | Normal-2435, Cancer-2511 | Kaggle |
| [10] | Histopathological | AlexNet, VGG-16, VGG-19, ResNet-50 | Normal-1656, Cancer-6665 | Private |
| [11] | Histopathological and Color image | ResNet50, MobileNetV2, VGG16, VGG19, DenseNet | Histo: Normal-2494, Cancer-2698 Color: Normal-44, Cancer-87 | Private |
| [13] | Color image | ResNet101, Faster R-CNN | Total-2155 | Private |
| [19] | Color image | AlexNet, GoogLeNET, VGG19, Inceptionv3, ResNet50, SqueezeNet | Total-200 | Private |
| [12] | Histopathological | ResNet, Inception Networks, U-Net | Total=143 | Private |
| **This study** | **Color image** | **All four variants of MobileNet (V1, V2, V3Small, V3Large), MobileViT-S** | **Total-296, Normal-148, Cancer-148** | **Private** |

et al. [13] MeMoSA project used ResNet-101 for image classification and Faster R-CNN for object detection, achieving promising F1 scores for the early detection of oral cancer using annotated clinical images and proposed to develop healthcare application. Although heavyweight architectures yield the best results, implementing them on mobile devices is challenging. These models can be integrated with webportal applications instead. For example, Cogan et al. [22] developed a mammogram screening telemedical webportal application using Faster R-CNN with ResNet-101, integrating a model with approximately 44 million parameters to classify breast cancer images.

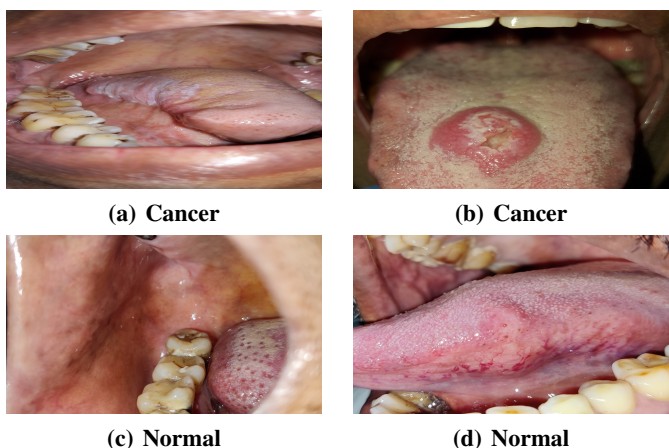

**(a) Cancer**  **(b) Cancer**

**(c) Normal**  **(d) Normal**

Fig. 2. Sample images from the SmartOralpix Dataset, which are captured using smartphones

### B. Lightweight Architecture

Howard et al. [5] developed MobileNetV1 architecture to accommodate constraints of mobile and embedded devices.

This research received increased attention due to its popularity of reduced memory requirement by neural network models known as lightweight models. These architectures are specifically designed to minimize computational requirements thus making them ideal for implementation in memory-constrained devices. The key idea is to use Depthwise Seperable convolution (depth-convolution followed by point-wise convolution) instead of standardized convolution, that results in less computation. The extensive research on this concept led to MobileNetV2. Sandler et al. [6] achieved much less computation requirement compared to MobileNetV1, by introducing linear bottlenecks and inverted residuals. This research contribution led to MobileNetV3 [7], which uses MobileNetV2 approach combined with squeeze and excitation networks.

Transformers have recently garnered significant attention due to the success of GPT models in natural language processing. Despite their exploration in image processing tasks, Vision Transformers are typically too computationally intensive for mobile devices. Mehta et al. [8] proposed a hybrid architecture combining the strength of CNNs and Transformers. This MobileViT architecture is designed to be lightweight and efficient, making it suitable for mobile and resource-constrained environments. Many deep learning-based automatic oral cancer detection systems exist. However, a lightweight model that is integrated into a healthcare app that is downloadable from playstore doesn't exist. To the best of our knowledge, a hybrid light weight architecture has not been explored in oral cancer research.

### III. MATERIALS AND METHODS

#### A. Database [SmartOralpix]:

A customized oral database called 'SmartOralpix' has been created by collecting images captured by dentists through smartphones. These are privately collected images. See Fig.2

for example images from the dataset. The sample size for each class in SmartOralpix is shown in Table II. Each image in SmartOralpix dataset has been carefully labeled by dentists/oral surgeons, ensuring accurate and reliable annotation. To overcome the problem of limited images we have, we applied on-the-fly data augmentation technique that enhanced the training image size to more than 128,000 which is described in the next section.

TABLE II
SMARTORALPIX DATABASE

| Class | No. of Images |
|---|---|
| Normal | 148 |
| Cancer | 148 |

### B. Datasets and Data augmentation

SmartOralpix is split into training data (80%) and validation data (20%). For testing, we use Oral Cancer dataset from Kaggle. The test dataset comprises of 21 cancerous images and 10 normal images. The test dataset should be from an independent source to bring out the clear generalization of the model and so we have relied on this test dataset even though its size is extremely small. Although the size is not good enough to provide a statistically significant result, it is sufficient for prototyping the model.

The training dataset being not sufficiently large enough has a high chance of overfitting. To address this challenge, data augmentation is employed to expand the scope of the existing dataset. We prefered on-the-fly data augmentation instead of pre-augmented dataset. This method introduces a degree of variability within training data, thus resulting in improved generalization.

The orientation of a lesion relative to the camera can indeed vary depending on how the photo was taken. However, it's also crucial to preserve the anatomical orientation when processing medical datasets. Implementing all possible augmentations, such as excessive rotations, shears, or flips, could potentially distort crucial anatomical details, leading to inaccuracies in diagnosis or assessments. So, we followed the safety of data augmentation standards proposed by Shorten et al. [23]. The following are the geometric transformations used. Rotation range=10°; width and height shift range=0.2; zoom and shear range=0.2; Horizontal flip is used, whereas Vertical flip is not used as it completely changes the anatomy.

### C. Data Preprocessing

We utilized models with pre-trained weights from the ImageNet database for our experiments. By adopting the same preprocessing steps used in the original ImageNet training, we ensured consistency and maximized the benefits of the transfer learning strategy. The preprocessing involves three steps: reading, resizing and scaling. The first step: OpenCV library is used, which loads the image in GBR (green/blue/red) then converted to RGB (red/green/blue). The second step: resizing the input image according to the architecture's input size.

All versions of MobileNet has an input size of 224×224×3, whereas MobileViT-S has an input size of 256×256×3. See Table III for details on the input dimensions for each of the architectures. The third step: scaling pixel values to [-1,1]. In all architectures except for the MobileNetV3, a pre-processing step to accomplish this is implemented explicitly, whereas MobileNetV3 architecture has a built-in function.

### D. Cross Domain Adaptation

Due to the limited size of our training dataset, we leverage the concept of transfer learning. All selected lightweight models have been pre-trained on the ImageNet database, which is not a medical database. This approach enables cross-domain adaptation, compensating for the absence of a domain-specific database or model tailored to our specific problem. We maintain the base model in a frozen state, adding a few layers on top that are exclusively trained on the SmartOralpix database. It's crucial to ensure that these newly added layers have adequately converged during training. This verification step must be completed before initiating the fine-tuning process of the pre-trained layers, laying the groundwork for a successful transfer learning strategy.

### E. Experimental Setup

All studies related to this research were carried out using TensorFlow (2.12.0 version) and TensorFlow Lite Support (0.4.4 version), with Python (3.8.19 version) on a desktop with MAC MINI. All training and testing were conducted on the GPU, utilizing Apple's Metal API for acceleration. The Mac Mini features 8GB RAM, an M1/M2 chip, and a 16-core Neural Engine. Android Studio 2022.3.1 platform is used for developing OCD (Oral Cancer Detector) mobile application.

## IV. PROPOSED METHOD

The proposed approach involves appending additional layers to the base model, with these layers being specifically tailored to the domain of interest. Given that this problem involves a cross-domain adaptation and our dataset is very small, incorporating domain-specific layers near the final network layer can significantly enhance the likelihood of achieving optimal results. This approach leverages the fine-tuning strategy, where the base model's pre-trained features are adapted to better suit the specific task at hand, thus maximizing performance despite the limited amount of data. We experimented with this transfer learning strategy using both CNN architectures (specifically all variants of MobileNet) and hybrid architecture (MobileViT-S). Finally, we compare the experimental results from all the models and picked the best model to implement it as an Android application. The research workflow for experimentation, evaluation, comparison of results, and integration into Android application are shown in Fig.1.

### A. Fine-tuning Layers

Initially, we augmented the pre-trained model by integrating an additional 15 layers specifically designed to capture the unique features of our dataset. By deeply customizing the

TABLE III
SUMMARY OF MOBILE-FRIENDLY MODELS UTILIZED IN THIS STUDY

| Models | Architecture | Input image size (W × H × D) | No.of parameters (millions) |
|---|---|---|---|
| MobileNetV1 [5] | CNN | 224 × 224 × 3 | 4.2 |
| MobileNetV2 [6] | CNN | 224 × 224 × 3 | 3.4 |
| MobileNetV3Small [7] | CNN | 224 × 224 × 3 | 2.5 |
| MobileNetV3Large [7] | CNN | 224 × 224 × 3 | 5.4 |
| MobileViT-S [8] | CNN+Transformer | 256 × 256 × 3 | 5.6 |

model beyond adding a classification layer for the target task and fine-tuning the existing layers [20], we enhanced the integration of domain-specific knowledge, thereby improving the model's ability to manage the distinctive characteristics of our data. We first trained on our newly added layers using a moderate learning rate of 0.001. This rate was carefully chosen to allow these layers to adapt quickly to the new data without altering, keeping the features in the base model frozen. These newly added layers use Swish activation function [24] which is proved to enhance performance in deeper network layers. Once these layers had trained well and converged effectively, demonstrating stability and improved performance, we proceeded to unfreeze the base layers. This allowed us to fine-tune the entire model, including both the newly added and base layers, at a lower learning rate below 0.001. This fine-tuning step ensures that the entire model adjusts harmoniously to the dataset, enhancing overall accuracy and robustness.

### B. Width Multiplier

The width multiplier, denoted by alpha is a global hyperparameter that is used to built computationally efficient models. Its value lies between 0 and 1 [5], [6], [7]. After conducting various empirical experiments, we determined that a width multiplier of 1 is sufficient for all MobileNet variants in our research problem. This configuration strikes a balance between model complexity and performance, providing adequate capacity to learn the necessary features without overfitting, especially given our small training dataset.

### C. Manual Leaning Rate scheduler

In our training process, we employed a manual learning rate adjustment strategy, guided by continuous monitoring of accuracy and loss graphs over epochs via TensorBoard. An integral part of this strategy was the decision to manually tweak the hyperparameters and learning rates by observing model performance. This allowed for precise adjustments, selecting the crucial learning rate for fine-tuning, instead of relying on an automatic learning rate scheduler. This criterion was set to ensure that the model had reached its optimal performance state under the current configuration before any further adjustments were deemed necessary.

### D. Optimizing for Mobile Application

Given that MobileNetV3Large emerged as the best-performing model, we used the TensorFlow Lite (TFLite) converter to transform the TensorFlow model into TFLite format. This conversion was essential to optimize the model

for mobile devices, ensuring efficient on-device inference. The TFLite converter allows us to maintain the performance of the original TensorFlow model while producing a lightweight and fast version suitable for mobile deployment.

## V. RESULTS AND DISCUSSION

During training, we used model checkpoints to save the best weights at periods of minimum validation loss, ensuring effective generalization of input image data. We selected binary cross-entropy as the loss function and Adaptive Moment Estimator (Adam) as the optimizer. A batch size of 32 works well based on experimentation, as our dataset performed poorly with higher batch sizes.

### A. Evaluation Metrics

It is important to evaluate the model to understand its efficiency and accuracy. Various metrics, such as the Confusion matrix, F1 score, Accuracy and AUC-ROC (Area Under the Receiver Operating Characteristics Curve) are used for this purpose. In our confusion matrix, we label '0' for normal (negative class) and '1' for cancer (positive class). The positive class is set to cancer because it is critical from a medical perspective to minimize false negatives (i.e., misclassifying cancer as normal). Misclassifying a negative instance as positive is less harmful compared to the risk of missing a true positive (cancer case). In the context of cancer screening, sensitivity (also known as recall) is more important than specificity. Sensitivity measures the proportion of actual positives (cancer cases) correctly identified by the model, which is crucial to ensure that cancer cases are not missed. High sensitivity reduces the risk of false negatives, which is vital in medical diagnosis to ensure timely treatment and intervention. Specificity, on the other hand, measures the proportion of actual negatives (normal cases) correctly identified. While high specificity is also desirable, it is less critical in the context of screening compared to sensitivity, as the consequences of a false negative (missing a cancer diagnosis) is more severe than a false positive (misidentifying a normal case as cancer).

In Fig. 3, we present the confusion matrices, accuracy curves, and loss curves for each of the experimented models. In Fig. 4, we plot the Receiver Operating Characteristic (ROC) curves which shows the performance exceeding 0.5 for all models. It is important to note that direct comparisons with other studies are not feasible because the dataset used in this research is private.

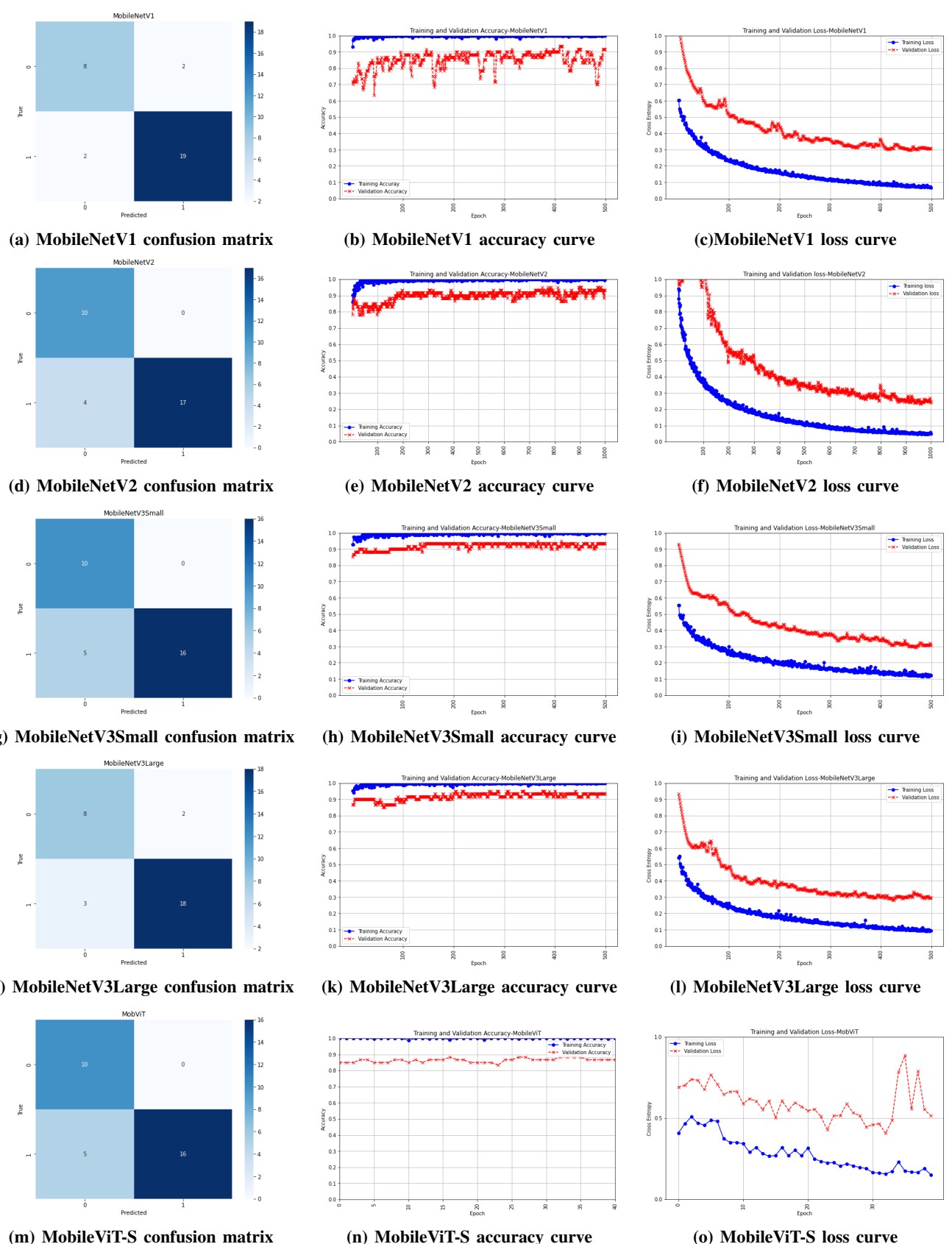

Fig. 3. The Confusion Matrices, Accuracy curves, and Loss curves are displayed for each lightweight model predicting oral cancer. In the confusion matrix, the label 0 represents Normal class, while the label 1 represents Cancer class.

| Trained Models | Precision(%) | Sensitivity(%) | Specificity(%) | F1 score(%) | Accuracy(%) | AUC | Inference time(sec) |
|---|---|---|---|---|---|---|---|
| MobileNetV1 | 90 | 90 | 80 | 90 | 87 | 0.90 | 0.0288 |
| MobileNetV2 | 100 | 81 | 100 | 89 | 87 | 0.92 | 0.0400 |
| MobileNetV3Small | 100 | 76 | 100 | 86 | 84 | 0.92 | 0.0059 |
| **MobileNetV3Large** | **90** | **86** | **80** | **88** | **84** | **0.92** | **0.0171** |
| MobileViT-S | 100 | 76 | 100 | 86 | 84 | 0.92 | 0.0295 |
| *Dentist's score* | *100* | *90.5* | *100* | *95* | *93.5* | *-* | *-* |

## B. Result and Analysis

The performance metrics from the model experiments are summarized in Table IV. The analysis reveals that Mo-bileNetV1 excels with good scores across all metrics: precision of 90%, specificity of 80%, sensitivity of 90%, F1 score of 90%, and accuracy of 87%. However, the continuous fluctuations in validation accuracy, even after training for 400 epochs, highlights the model's inconsistency and instability. Due to these observed fluctuations, MobileNetV1 was not selected. MobileNetV3Large and MobileNetV2 are the best performers among stable models. When comparing these two models, MobileNetV3Large demonstrates higher sensitivity of 86% and a more stable learning curve, which suggests better generalization. Also, the MobileNetV3Large converged faster when compared to MobileNetV2. Moreover, based on Fig 4, MobileNetV3Large demonstrates a more stable ROC curve compared to MobileNetV2. When comparing the inference time of all the TFLite models, MobileNetV3Small emerges as the fastest, followed by MobileNetV3Large. However, MobileNetV3Large offers more stable performance metrics across various tasks, making it a dependable option for mobile applications despite its slightly longer inference time. As a result, MobileNetV3Large TFLite model has been integrated into the OCD application, that is being developed.

Additionally, MobileNetV3Small and MobileViT-S exhibit a significant drop in sensitivity, adversely affecting their F1 scores and overall accuracy. We expected MobileViT-S to perform well, given its advanced architecture designed for vision tasks. It exhibits significantly lower sensitivity of 76%, which severely affects its reliability, despite having a high specificity.

## VI. MOBILE APPLICATION - ANDROID

We named this Android application as Oral Cancer Detector (OCD). The OCD encompasses five distinct steps. The term "step" in this context pertains to a distinct screen that becomes visible upon button interaction. The first step features an OCD splash screen containing a disclaimer. Proceeding to the second step, users are presented with a choice: either to instantaneously capture a live photo using their device's camera ("Take Photo" button) or to peruse their image gallery ("Upload" button) and pick an image. In the third step, we suggest using "Crop" button, to crop the image and focus on the lesions. Once the image is ready, users can move forward by clicking on the "Analyze" button, which is the fourth step. This button smoothly integrates with our TFLite

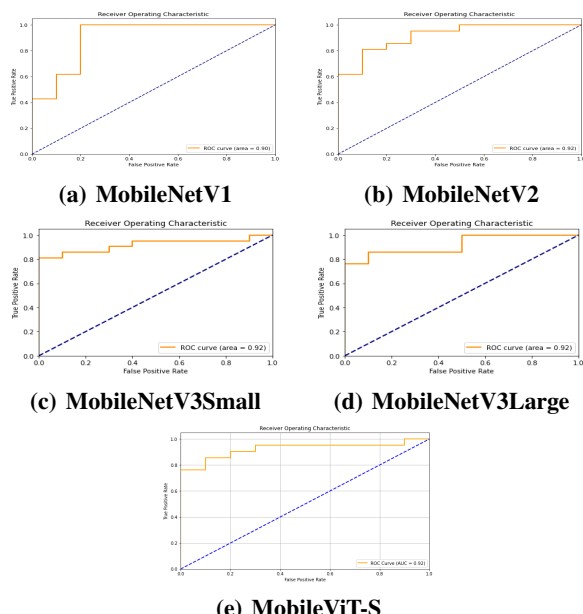

**(a) MobileNetV1**    **(b) MobileNetV2**

**(c) MobileNetV3Small**    **(d) MobileNetV3Large**

**(e) MobileViT-S**

Fig. 4. The solid line shows the Receiver Operating Characteristic curve (ROC curve: sensitivity versus 1-specificity) obtained for the Kaggle dataset. The AUC values are determined from these plots.

model. Upon selection, the model pre-processes the input image followed by prediction and displays the result in the fifth activity. If the application interprets the lesion to be cancerous, then this step will display "Referral Suggested", if not cancerous, it will display "No Referral Suggested". The requirement of this application is that the suspicious lesion should be cropped properly, to obtain the accurate result. Fig. 5 shows the screenshots of the user interface in our OCD android application.

## VII. CONCLUSIONS

In our experiments, MobileNetV3Large performed exceptionally well with accuracy of 84% and sensitivity of 86%, even with a relatively small unaugmented training data and its simple network architecture. This performance is notable, as it indicates that MobileNetV3Large can provide accurate results without requiring extensive computational resources or large amounts of unaugmented training data. Although other models did not show balanced performance, we believe that with more unaugmented training data and further fine-tuning strategy, their performance could be significantly improved. We had anticipated that MobileViT-S would outperform other models

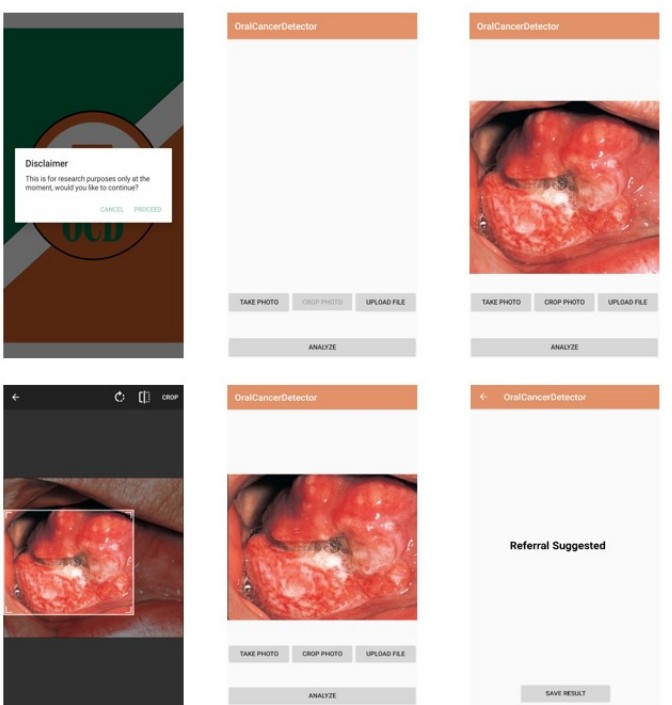

Fig. 5. Screenshots of User Interface (UI) in our OCD Android application.

due to its hybrid approach. However, this was not observed in our initial results. We attribute this to the limited size of our unaugmented training dataset and the specific nature of our task. With additional unaugmented training data and further optimization, we expect MobileViT-S performance to improve and potentially even surpass that of all variants of MobileNet.

## ACKNOWLEDGMENT

Lakshman Tamil expresses his gratitude for the inspiration provided by the late Mr. Gregory Stubblefield for this research. He dedicates this research paper to him. ChatGPT-4.0 was used to correct the English in some parts of this paper.

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
