# OpenReview forum: "Oral Cancer Detection using Mobile Vision Technology"
_IEEE.org/EMBS/BHI/2024/Conference — IEEE BHI'24_

### Official Review · Reviewer_yKcJ · 2024-08-10
**A mobile application using lightweight deep learning models for early detection of oral cancer, optimized for accessibility and on-device processing.**

**Overall Rating:** 7
**Confidence:** 4

**Other Quality Metrics:**

Clarity of Writing: Good
Clinical Significance: Great
Methodological Novelty: Good
Experiments and Results: Fair

**Questions For The Authors:**

1. How do you anticipate your model will perform on datasets from different populations or regions, particularly given the limited size and scope of the SmartOralpix dataset? Have you considered testing the model on more diverse datasets to evaluate its generalizability?
2. Have you tested your model on any publicly available datasets or in collaboration with external institutions to validate its performance? If not, do you plan to conduct such tests to strengthen the evidence of your model’s effectiveness?
3. Can you elaborate on how the user experience and interface design were considered in the development of the OCD application? Were any usability studies conducted to ensure the app is accessible and easy to use for individuals with varying levels of tech literacy?
4. While the paper emphasizes the importance of sensitivity in detecting oral cancer, how do you address the trade-offs with other performance metrics like specificity and precision? How do you ensure that the model maintains a balance between minimizing false negatives and avoiding too many false positives?

**Strengths:**

1. The paper demonstrates a creative application of mobile vision technology to address a critical healthcare challenge—early detection of oral cancer. By leveraging smartphones, which are widely accessible, the study presents a cost-effective and scalable solution that has the potential to significantly improve early cancer detection, especially in resource-limited settings.
2. One of the key strengths is the focus on creating a mobile application that operates entirely on-device, ensuring privacy and accessibility even in areas with limited or no internet connectivity. This approach makes the technology highly accessible to underserved populations, who might not have regular access to healthcare facilities.
3. The authors conducted a thorough evaluation of several lightweight deep learning models, including different variants of MobileNet and a hybrid CNN-Transformer model (MobileViT-S). This comparative analysis allows for the identification of the most suitable model (MobileNetV3Large) for the specific task, balancing performance with computational efficiency, which is crucial for mobile applications.
4. The successful development and deployment of a working mobile application (Oral Cancer Detector) highlight the practical relevance of the research. The app's ability to provide immediate, on-device analysis and feedback is a significant achievement, demonstrating the real-world applicability of the technology.
5. The paper effectively addresses the challenge of a small dataset by employing transfer learning and data augmentation techniques. This approach not only enhances the model's performance but also illustrates the potential of transfer learning in medical image analysis, where labeled data can be scarce.

**Summary Of The Paper:**

The paper explores the development and application of mobile vision technology for early detection of oral cancer using a smartphone-based telemedical application. Oral cancer is a significant global health issue, particularly in middle-income countries, where access to regular, affordable dental check-ups is limited. Early detection is crucial for improving survival rates, but the high costs of traditional screening methods pose a barrier to many individuals. This research addresses this gap by converting smartphones into cancer screening tools through the use of deep learning models that can analyze images of oral lesions captured by the device's camera. The paper details the process of creating a mobile application that utilizes deep learning algorithms to classify images of oral lesions, identifying potential cancerous growths. The authors developed a customized database, SmartOralpix, consisting of images captured by dentists, which was used to train and test the deep learning models. Given the limited size of this dataset, transfer learning and data augmentation techniques were employed to enhance the model’s performance.

Several lightweight deep learning models were explored, including various versions of MobileNet (V1, V2, V3Small, and V3Large) and MobileViT-S, a hybrid CNN-Transformer model. These models were selected due to their suitability for mobile applications, given their lower computational requirements compared to more complex models. The research involved fine-tuning pre-trained models from the ImageNet database, adapting them to the specific task of oral lesion classification. The models were evaluated based on metrics such as accuracy, sensitivity, specificity, and F1 score, with MobileNetV3Large emerging as the best-performing model. This model achieved an accuracy of 84%, sensitivity of 86%, and an F1 score of 88%, making it the preferred choice for integration into the mobile application.

The development of the mobile application, named Oral Cancer Detector (OCD), is a key outcome of this research. The application allows users to capture or upload images of oral lesions, which are then analyzed by the integrated deep learning model to determine the likelihood of cancer. The app operates entirely on-device, ensuring privacy and accessibility, particularly in remote or resource-limited areas where internet connectivity may be unreliable. The app provides immediate feedback, indicating whether a referral to a healthcare professional is suggested based on the analysis. The paper discusses the steps involved in optimizing the model for mobile deployment, including the use of TensorFlow Lite to convert the model into a format suitable for efficient on-device inference.

The results of the experiments showed that while MobileNetV3Large provided the best overall performance, the other models also demonstrated potential, though they were less consistent or had lower sensitivity. The study highlights the importance of sensitivity in medical diagnostics, emphasizing the need to minimize false negatives to ensure that potential cases of cancer are not missed. The paper concludes by noting the limitations of the research, particularly the small size of the training dataset, which likely impacted the performance of the more complex models like MobileViT-S. The authors suggest that with a larger, unaugmented dataset, the performance of these models could be significantly improved.

This research represents a significant step forward in making oral cancer screening more accessible and affordable through the use of mobile technology. By integrating deep learning models into a user-friendly mobile application, the study demonstrates the potential of smartphones to serve as powerful tools for early cancer detection, particularly in underserved populations. The success of this approach also opens the door to similar applications in other areas of healthcare, where mobile technology can be leveraged to bridge the gap between patients and early diagnostic services.

**Weaknesses:**

1. The study relies on a relatively small dataset (SmartOralpix) for training and testing the deep learning models. The limited size of the dataset may have constrained the model's ability to generalize well, particularly when applied to more diverse or larger populations. This also likely impacted the performance of more complex models like MobileViT-S.
2. The model's performance is primarily evaluated using a private dataset, with limited external validation on publicly available datasets. This raises concerns about the generalizability of the model to other populations and settings, which is crucial for ensuring the reliability of the technology in real-world applications.
3. While the paper discusses the development of a mobile application, there is no mention of real-world testing or deployment of the app with actual users. Real-world testing is essential to understand how the application performs in practice, particularly in varying lighting conditions, different smartphone models, and among users with varying levels of tech literacy.
4. Given the small size of the training dataset, there is a risk of overfitting, where the model performs well on the training data but may struggle with unseen data. Although transfer learning and data augmentation were used to mitigate this, the paper does not discuss in depth the steps taken to ensure the model's robustness against overfitting.
5. Although the paper evaluates several lightweight models, the exploration of more advanced models, such as Vision Transformers (beyond MobileViT-S), is limited. These models, though computationally intensive, could offer better performance if properly adapted and optimized for mobile devices.

---

### Official Review · Reviewer_zWg4 · 2024-08-12
**Oral Cancer Detection using Mobile Vision Technology - Review**

**Overall Rating:** 6
**Confidence:** 5

**Other Quality Metrics:**

(a) Clarity of writing - great
(b) Clinical Significance - good
(c) Methodological Novelty - good
(d) Experiments and Results - fair

**Questions For The Authors:**

1. How do you make sure the data characteristics from your training data (SmartOralpix) and test data (Kaggle) have aligned distribution characteristics?

2. All the models used in this study are pre-trained models (trained from ImageNet), should we have another base-line (eg. small ResNet-18) and train from scratch?

Similar to the question (1), there will be a domain shift between trained data (ImageNet), and test data (fine-tuning the last layer in this case on clinical domain data), please provide clear justification on that.

3. The final reports in Table IV is sufficient for comparison between models' performance. However, we are dealing with very limited data availability. Should we have at least k-fold cross validation, or run multiple times (eg. 5 times) to have a statistical results for each model?

4. In case of misclassification (especially false negative cases), should we have any analysis to see how the model fail on the predictions, based on that we can explore the interpretability and explainability of the model?

**Strengths:**

This work is a comparative analysis of the lightweight framework to work on-divide decision support systems.
The idea is promising when we can use pre-trained models for clinical applications on the device.

**Summary Of The Paper:**

The paper discusses comparative experiments with lightweight machine learning models, highlighting MobileNetV3Large, which achieved notable performance metrics (84% accuracy, 86% sensitivity, and 80% specificity) on test data. This approach emphasizes efficient, private, and accessible screening capabilities.

**Weaknesses:**

The only consideration is how to justify the training data, and test data does not have many discrepancies in data characteristics.
Additionally, it should have a small model training from scratch as a baseline to compare with using pre-trained models.

---

### Official Review · Reviewer_s3WH · 2024-08-17
**Practical solution and approach, requires much more detail, comparisons for supporting novelty**

**Overall Rating:** 5
**Confidence:** 4

**Other Quality Metrics:**

(a) Clarity of writing;  good
(b) Clinical Significance; great
(c) Methodological Novelty; fair
(d) Experiments and Results: good

**Questions For The Authors:**

1. How does the model's performance compare to expert clinicians on the same test set?
2. What steps were taken to ensure patient privacy and consent in creating the custom dataset?
3. How might the app's performance be affected by varying image quality from different mobile devices?
4. Although the paper talks about computational efficiency in using MobileNetV3Large, the computational time / power for inferences are not compared and hence the argument is not supported strongly.

Additionally:
1. Sample images from the private dataset will help understand how these images look like - and marking the cancer / non-cancerous one in those images will help further understand the complexity of the problem.
2. It seems that the very small test dataset limits statistical significance of results in terms of generalizability - although safe augmentation is utilized. It would be better if the validation sensitivity and test sensitivity are compared to provide an idea about generalizability.

**Strengths:**

1. Addresses an important healthcare need with a practical mobile solution
2. Evaluates multiple lightweight architectures suitable for mobile deployment
3. Uses transfer learning and data augmentation to overcome limited training data
4. Provides detailed methodology and performance metrics for model comparison
5. Implements the model in a functional Android application

The authors should try to translate this into clinical practise, which will have great impact on patient care.

**Summary Of The Paper:**

This paper presents research on developing a mobile application for oral cancer detection using deep learning. The authors created a custom dataset of oral images, experimented with various lightweight convolutional neural network architectures suitable for mobile devices, and implemented the best performing model in an Android app. The app allows users to take or upload photos of oral lesions for instant on-device screening.

**Weaknesses:**

Although the authors have presented a practical solution for this problem, it would be better if more details are provided that will help evaluate the paper better.

1. Sample images from the private dataset will help understand how these images look like - and marking the cancer / non-cancerous one in those images will help further understand the complexity of the problem.
2. It seems that the very small test dataset limits statistical significance of results in terms of generalizability - although safe augmentation is utilized. It would be better if the validation sensitivity and test sensitivity are compared to provide an idea about generalizability.
3. The test images and train image variability is not mentioned, which makes it difficult to understand how divergent they were.
4. Lacks comparison to expert human performance as a baseline
5. How many dentists / doctors labelled these train images ? WHat was the confidence in labelling?
6. No discussion of potential ethical issues or regulatory requirements
7. Limited explanation of the customized database creation process
8. No user testing or evaluation of the mobile app's usability
9. Although the paper talks about computational efficiency in using MobileNetV3Large, the computational time / power for inferences are not compared and hence the argument is not supported.

A major drawback is that there are similar works [13] and [19] that the paper has cited. However, a strong comparison with those are missing.

---

### Decision · Program_Chairs · 2024-09-23

Accept